# Influence of Intravitreal Therapy on Choroidal Thickness in Patients with Diabetic Macular Edema

**DOI:** 10.3390/jcm12010348

**Published:** 2023-01-01

**Authors:** Patricia Udaondo Mirete, Carmen Muñoz-Morata, César Albarrán-Diego, Enrique España-Gregori

**Affiliations:** 1Department of Ophthalmology, Hospital Universitario y Politécnico La Fe, 46026 Valencia, Spain; 2Aiken Clinic, 46004 Valencia, Spain; 3Department of Surgery, Universidad de Valencia, 46010 Valencia, Spain

**Keywords:** anti-VEGF, diabetic macular edema, dexamethasone implant, intravitreal therapy, OCT, aflibercept, ranibizumab, choroid

## Abstract

Objective: This study aimed to analyze the variation in subfoveal choroidal thickness (SFCT) and its relationship with the variation in central macular thickness (CME) in response to intravitreal therapy with an antiangiogenic (anti-VEGF) drug or corticosteroid in type 2 diabetic patients with diabetic macular edema (DME). Material and methods: This retrospective study included 70 eyes of 35 patients: 26 eyes received 4−5 intravitreal injections of aflibercept, 26 eyes were treated with a single intravitreal implant injection of dexamethasone, and 18 eyes without DME did not receive intravitreal therapy. SPECTRALIS® optical coherence tomography (OCT) (Heidelberg Engineering, Heidelberg, Germany) was used to measure the SFCT and CME before and at the end of the follow-up period. Results: The mean reductions in CME were 18.8 +/− 14.7% (aflibercept) and 29.7 +/− 16.9% (dexamethasone). The mean reductions in SFCT were 13.8 +/− 13.1% (aflibercept) and 19.5 +/− 9.6% (dexamethasone). The lowering effects of both parameters were significantly greater in the group treated with the dexamethasone implant (*p* = 0.022 and *p* = 0.046 for CMT and SFCT, respectively). Both therapies significantly decreased both CME and SFCT, independent of factors such as age, sex, previous intravitreal therapy, antidiabetic treatment, and the time of diabetes progression. There were no changes in the mean values of CME and SFCT in the untreated eyes. Conclusions: SFCT significantly decreased in response to intravitreal therapy with anti-VEGF or corticosteroids, irrespective of age, sex, previous intravitreal therapy, antidiabetic treatment, and the time of diabetes progression. There was a correlation between the changes in CME and SFCT after intravitreal therapy with aflibercept or dexamethasone implantation. SFCT was not a good predictor of the CME response but could be used to monitor the response to treatment. Local intravitreal therapy only affected the treated eye.

## 1. Introduction

Diabetes mellitus (DM) is a major public health problem. This is due to the high prevalence rate of diabetes mellitus (DM), the severe health complications DM can cause, and the resulting increase in healthcare costs. The retina is the body structure that most frequently presents microvascular alterations in DM [1]. Of the 285 million people who have diabetes worldwide, 25–30% suffer from some degree of diabetic retinopathy (DR) [1,2,3].

Diabetic macular edema (DME) is the main cause of blindness in diabetic patients of working age [4]. Up to 10% of diabetic patients have clinically significant macular edema (CSME). 

The main risk factors associated with the development of DME are the duration of diabetes, poor glycemic control, and hypertension, with Afro-American heritage and dyslipidemia being potentiating factors [5].

Sustained hyperglycemia activates a number of intracellular metabolic pathways, including the hexosamine and polyol pathways, C-reactive protein, and free radical formation. Crucial elements in the development of DME are glycation end products and increased levels of inflammatory mediators such as cytokines, chemokines, prostaglandins, adhesion molecules, and matrix metalloproteinases. Leukocyte chemotaxis contributes to the disruption of the BHR, leading to endothelial dysfunction and decreased vascular perfusion [4,5,6,7]. There is free-radical-induced and glycation end product induced neurodegeneration that also contributes to the pathogenesis of DME [7].

Optical coherence tomography (OCT) has become the gold standard test for the diagnosis and follow-up of DME [8,9] and over the years has led to new classifications based on the structural changes found by imaging. One of the most complete and current is the classification by Panozzo et al. [10], which divides DME according to various characteristics such as the edema morphology and vitreoretinal interface status.

On the other hand, based on the response to intravitreal therapy with vascular endothelial factor (VEGF) inhibitors or corticosteroids, numerous articles have been published describing OCT imaging biomarkers that are predictive of responses or prognostic to improve the management of patients with DME [11,12,13].

The thickness of the choroid in a healthy eye is variable depending on the location, with a minimum value of 100 µm in the ora serrata and a maximum value of 250–300 µm at the level of the fovea. Histological examinations have revealed that in diabetic patients the choroid undergoes pathological changes such as increased tortuosity of the vessels and the appearance of areas of fibrosis [14,15]. Metabolic changes that affect the retina also affect the choroid [16,17].

One parameter that is studied in relation to DME is the subfoveal choroidal thickness (SFCT), although results have been inconsistent and even contradictory. Some studies have concluded that DME correlates with a thinning of the GCSF [18,19,20,21], while other studies report increases in choroidal thickness [22,23]. This discrepancy may be explained by the wide spectrum of pathological changes that occur in DR. Some authors describe an association between SFCT and the severity of DR, with increased choroidal thicknesses in more severe stages and no correlation with the presence of DME [18].

SFCT is a parameter for which no consistent results have been obtained with regards to its relationship with DME and the response of both to intravitreal therapy. The present work studies changes in choroidal thickness and its relationship with DME in response to intravitreal therapy in order to assess its usefulness as a biomarker.

The main objective is to study SFCT variation and its correlation with CMT variation in response to intravitreal therapy with anti-VEGF or corticosteroids in type 2 diabetic patients with DME.

Secondary objectives:To demonstrate whether there are differences in the variation in macular and choroidal thicknesses in patients treated with anti-VEGF compared to patients treated with corticosteroids.To analyze whether there is a variation in CMT and GCSF in untreated eyes after intravitreal treatment in the contralateral eyes.To evaluate the influence of the following variables on the response to intravitreal therapy with anti-VEGF or corticosteroids:-Age;-Gender;-Previous intravitreal therapy;-Type of antidiabetic treatment;-Time course of type 2 diabetes.

## 2. Material and Methods

### 2.1. Design

This was a retrospective study that was conducted in patients with type 2 DM with subfoveal macular edema who were treated with intravitreal therapy, either anti-VEGF or corticoid, in the Ophthalmology Department of the Hospital Universitario y Politécnico la Fe in Valencia from January 2020 to July 2021.

### 2.2. Participants and Methods

In total, 70 eyes of 35 type 2 diabetic patients were included, 52 eyes with DME (treatment group) and 18 eyes without edema (control group); of the 52 eyes with DME, 26 eyes were treated with anti-VEGF (aflibercept) and 26 eyes were treated with corticosteroids (dexamethasone implant).

The inclusion criteria for the treatment group included patients over 18 years of age diagnosed with type 2 DM with DME in at least one eye treated by intravitreal therapy with anti-VEGF or corticosteroids with a minimum follow-up of 2 months in the corticosteroid group and 6 months in the anti-VEGF group from the first intravitreal injection and all eyes with availability of volume scan (high-speed resolution and 20º × 20º scan pattern), with follow-up performed in the same OCT SPECTRALIS® (Heidelberg Engineering, Heidelberg, Germany) and sufficient quality to measure morphological changes in the retina and choroid at baseline and at the end of the study period.

The measured parameters were the central macular thickness (CMT) and subfoveal choroidal thickness (SFCT), and all tomographic parameters were analyzed by a single observer (CMM). To measure the central macular thickness (CMT), the retinal thickness map analysis protocol was selected. On the other side, the SFCT was measured manually using the fundus image analysis window and the manual measurement tool; three measurements of the vertical distance between the outer boundary of the hyper-reflective line of the RPE and the sclerochoroidal interface at the subfoveal level were taken, and the mean was calculated (Figure 1).

### 2.3. Statistical Analysis

The main variables included the intravitreal treatment and the pretreatment and post-treatment CMT and SFCT.

Other variables included age, gender, previous intravitreal therapy (none, aflibercept, ranibizumab, or dexamethasone implant), the type of systemic antidiabetic treatment (insulin, oral antidiabetics, subcutaneously administered type 1 glucagon-like peptide analogues, or a combination of both), and the time course of type 2 diabetes (years).

The Sigmaplot v14 software for Windows (Systat Software, Inc., California, CA, USA) was used for the statistical analysis. For descriptive statistics, the arithmetic mean was used as the index of central tendency, and the standard deviation and the range, with its maximum and minimum values, were used as indices of dispersion. The level of statistical significance was set at 5% (*p* = 0.05).

The Shapiro–Wilk normality test was used to check the normality of the distributions of the studied variables. If the test was passed, parametric tests were used for inferential statistics, and if normality could not be assumed, the corresponding non-parametric tests were used.

The correlation between variables was studied by means of a linear regression and the calculation of the coefficient of determination, R2, which accompanies the regression equations and provides information on the goodness of fit of a model to the correlation of the variables it seeks to explain. A good correlation is considered to exist with R2 values above 0.85, and a very good correlation exists above 0.95.

#### 2.3.1. Within-Group Analysis

Within each treatment group, the paired-measures *t*-test was used to study possible differences between the values of the variables under study before and after treatment. The corresponding non-parametric test, the Wilcoxon rank test, was used if the data did not pass the normality test. To determine differences in sex distribution or previous intravitreal therapy within each treatment group, Fisher’s exact test was used.

The chi-square test was used to compare the variables prior to the intravitreal treatment. Since the chi-square test may not be sufficiently robust due to the sample size and the stratification into several pretreatment groups, the variables were categorized as follows to reapply the chi-square test:-The variable “Pretreatment” was dichotomized into yes (there was pretreatment) and no (there was not) groups.-The variable “Thinning in CMT or SFCT after treatment” was categorized into three levels:▪50 μm;▪50–100 μm;▪>100 μm.

The chi-square test was used to study the possible influence of the antidiabetic treatments (insulin, non-insulin antidiabetics, or a combination of both) on the improvement (thinning) in SFCT and CMT after intravitreal treatment in each of the two treatment groups. The variables were categorized as follows:-The variable “Antidiabetic treatment” was categorized into three levels:▪INS (insulin);▪ANI (non-insulin antidiabetic);▪COM (combined treatment with insulin and non-insulin antidiabetics).

#### 2.3.2. Comparison of Treatment Groups

To determine whether the populations of both treatment groups were homogeneous and that no biases were introduced that could invalidate the response analysis, the following statistical tests were employed: *t*-test (age), Fisher’s exact test (sex and previous intravitreal treatment), chi-square (systemic treatment for diabetes), and the Mann–Whitney test (time course of diabetes).

The Mann–Whitney test was used to detect pre- and post-treatment differences in SFCT or CMT between the two intravitreal treatment groups.

#### 2.3.3. Comparison with Untreated Oleander Eyes

The non-parametric Kruskall–Wallis test (non-parametric test corresponding to the one-way ANOVA) was employed to detect statistically significant differences in CMT or SFCT before and after treatment between the aflibercept, dexamethasone, and untreated oleander eye groups. Dunn’s multiple comparisons post hoc test was used to locate these differences between groups in the case of significant differences in the Kruskall–Wallis test.

## 3. Results

### 3.1. Aflibercept-Treated Group

Of the 26 eyes that were included in this group, 12 eyes (46%) had received some previous intravitreal treatment, while 14 eyes (54%) had not been treated before (Fisher’s exact test, *p* = 0.782).

#### 3.1.1. Differences in CMT with Treatment

The mean CMT went from 441.4 ± 75.9 μm to 357.8 ± 87.2 μm after treatment (Figure 2). The mean percentage reduction in CMT after aflibercept treatment was 18.8 ± 14.7%.

The CMT data for the aflibercept-treated group passed the normality test (Shapiro–Wilk, *p* = 0.920). A paired *t*-test found statistically significant differences between the pre- and post-treatment values (*t* = 6.526, *p* < 0.001) (Table 1).

When applying the chi-square test on these data, no relationship was found between having received pretreatment and improvement or thinning in CMT after intravitreal treatment with aflibercept (Chi^2^ = 2.465, *p* = 0.292).

The time of diagnosis of diabetes was also not an important factor in CMT reduction, although there was a slight trend towards greater improvements in eyes with less time of diabetes evolution.

There was no difference in CMT changes according to the type of systemic treatment (chi-square, Chi^2^ = 2.969, *p* = 0.563).

#### 3.1.2. Differences in SFCT with Treatment

The mean SFCT went from 274.7 ± 88.8 μm to 235.6 ± 83.4 μm after treatment. The mean reduction in SFCT after aflibercept treatment was 13.8 ± 13.1%.

The SFCT data for the aflibercept-treated group passed the normality test (Shapiro–Wilk, *p* = 0.756). A paired *t*-test found statistically significant differences between the pre- and post-treatment SFCT (*t* = 5.183, *p* < 0.001). The correlation between the SFCT before and after aflibercept treatment was high and statistically significant, with an R2 value of 0.814 (Table 1).

When applying the chi-square test on the categorized data, no relationship was found between the change in SFCT and having received previous intravitreal treatment (Chi2 = 1.413, *p* = 0.493). The change in SFCT was also independent of the time of diabetes progression and systemic treatment for DM (chi-square, Chi2 = 3.991, *p* = 0.407).

#### 3.1.3. Predictive Value of SFCT for CMT Response after Treatment

The regression line indicated a null usefulness of SFCT as a predictive biomarker of the CMT response after intravitreal treatment with aflibercept, with no correlation between the two variables and an R2 value of 0.02 (Figure 2).

### 3.2. Dexamethasone-Treated Group

This group consisted of 26 eyes belonging to 13 females (50%) and 13 males (50%). In this group, 16 eyes (62%) had received some previous intravitreal treatment, while 10 eyes (38%) had not been treated before.

#### 3.2.1. Differences in CMT with Treatment

The mean CMT went from 480.1 ± 97.3 μm to 328.0 ± 72.2 μm after treatment. The mean reduction in CMT after treatment with dexamethasone was 29.7 ± 16.9%.

The CMT data did not pass the normality test (Shapiro–Wilk, *p* < 0.05), so the non-parametric Wilcoxon rank test was used, which revealed statistically significant differences between CMT before and after treatment (Wilcoxon W= −351, *p* < 0.001) (Table 1). The median CMT before dexamethasone treatment was 496.0 m (Q1-Q3 interquartile range: 393.5 μm to 530.5 μm), while the median CMT after treatment was 320.0 μm (Q1-Q3 interquartile range: 285.3 m to 378.3 μm).

The CMT thinning achieved after dexamethasone treatment was not different between eyes with and without prior intravitreal treatment, both analyzing two groups separately (chi-square, Chi2 = 47.6, *p* = 0.407) and pooling both into one group (eyes with prior intravitreal treatment) and categorizing the variable CMT thinning after treatment into three levels: <50 μm, 50–100 μm, and >100 μm (chi-square, Chi2 = 0.914, *p* = 0.633).

As in the aflibercept group, the time of diabetes progression was not an important factor in the CMT change after dexamethasone treatment, although there was a slight trend towards greater improvements in eyes with less time of diabetes progression. There was no difference according to previous systemic treatment (chi-square, Chi2 = 5.709, *p* = 0.222).

#### 3.2.2. Differences in SFCT with Treatment

The mean SFCT went from 291.6 ± 87.1 μm to 229.8 ± 58.8 μm after treatment. The mean reduction in SFCT after treatment was 19.5 ± 9.6%.

The SFCT data passed the normality test (Shapiro–Wilk, *p* < 0.050), so the paired *t*-test was employed, which revealed statistically significant differences between the pre- and post-treatment values (*t* = 7.648, *p* < 0.001) (Table 1). In contrast to the CMT, there was a high correlation between the SFCT values before and after dexamethasone treatment, with an R2 value of 0.833.

As with the CMT changes, no difference in SFCT change was found between pretreated and naïve eyes (chi-square, Chi2 = 50.0, *p* = 0.394). The duration of diabetes and the systemic treatment received by the patients also did not influence the SFCT results, although there was a discrete tendency for greater post-treatment thinning in eyes with shorter durations of diabetes.

#### 3.2.3. Predictive Value of SFCT for Post-Treatment DME Response

The regression line indicated a very low usefulness of SFCT as a predictive biomarker of the CMT response after intravitreal treatment with dexamethasone, with an R2 value of 0.11 (Figure 3).

### 3.3. Comparison of Aflibercept and Dexamethasone Groups

Before starting treatment, there were no significant differences in the CMT (Mann–Whitney U = 257.5, *p* = 0.143) or in the SFCT (*t*-test *t* = 0.692, *p* = 0.492) between the two groups.

Therefore, both treatment groups were homogeneous in terms of age, sex distribution, previous intravitreal treatment, the systemic treatment of diabetes, years of diabetes evolution, and CMT and SFCT values before intravitreal therapy.

After intravitreal treatment, there were no differences in the CMT (Mann–Whitney U = 279.5, *p* = 0.288) or in the SFCT, (*t*-test *t* = −0.288, *p* = 0.774) between the two groups. Although the mean pre- and post-treatment values were similar in both groups, the weight loss in the CMT achieved with treatment was greater and was statistically significant in the dexamethasone-treated group (Mann–Whitney U = 212.0, *p* = 0.022). The improvement in the SFCT after treatment with dexamethasone was also significantly higher compared to treatment with aflibercept (*t*-test *t* = 2.050, *p* = 0.046) (Figure 4).

### 3.4. Control Group

In the untreated/control eye group, there were no significant differences in the mean SFCT or CMT values at baseline or at the end of the follow-up period. The mean reductions in CMT and SFCT were −1.7 ± 9.9% and −1.6 ± 11.3%, respectively. There was no reduction effect for both parameters in eyes not treated with intravitreal therapy.

## 4. Discussion

The present study evaluated and attempted to correlate the variation in choroidal thickness with the improvement in macular edema in response to intravitreal therapy with anti-VEGF (aflibercept) or corticosteroids (dexamethasone) in type 2 diabetic patients. Our results indicate that both intravitreal therapies significantly decrease both choroidal thickness and macular edema, independent of factors such as age, sex, previous intravitreal therapy, the systemic treatment of diabetes, and the time of diabetes evolution, and that the choroidal and macular thickness reduction effects were greater in the dexamethasone-treated group than in the aflibercept-treated group. The central macular thicknesses achieved in the dexamethasone-treated group were very similar to those in the group of eyes without macular edema. The subfoveal choroidal thicknesses (SFCT) of both treatment groups decreased to the same values as those of the oleophilic eyes without macular edema. A relevant finding of the present study is that baseline choroidal thicknesses (determined by the SFCT marker) are not predictive of the macular edema response to intravitreal treatment.

Monitoring the response to treatment using biomarkers is important for decision making. OCT allows the measurement of different anatomical parameters with high reproducibility to assess the degree of response to different treatments [24]. The choroid plays an integral role in metabolic support and outer retinal function [25,26]. Previous studies have shown that metabolic changes in diabetes primarily affect the choroid [27,28,29]. The SFCT was previously studied in the eyes of diabetic patients, and attempts have been made to correlate it with the existence of DME and with the response of the latter to different forms of treatment, without obtaining consistent results. On one hand, there are studies that relate the presence of DME to lower choroidal thicknesses, suggesting an ischemic origin for both findings (vascular or ischemic hypothesis) [18,19,21,30]. On the other hand, there are studies that have found increased choroidal thicknesses in eyes with DME, pointing to an inflammatory mechanism mediated by VEGF and other cytokines that are increased in the eyes of diabetic patients (inflammatory hypothesis) [12,22,23,31].

The decrease in choroidal thickness after intravitreal therapy demonstrated in the present study is consistent with previously published results. Campos et al. analyzed 126 eyes with newly diagnosed DME and demonstrated a significant reduction in SFCT after therapy with loading doses of aflibercept or ranibizumab for 3 months [32]. Likewise, Yiu et al. [33] and other authors have reported similar findings showing choroidal thickness thinning after anti-VEGF treatment [34,35]. Similarly, several studies with intravitreal corticosteroids have concluded that there are significant decreases in SFCT after treatment [36,37,38,39].

The present study shows a correlation between the improvement in macular edema and the decrease in central choroidal thickness, as determined by the SFCT in response to intravitreal therapy with anti-VEGF or corticosteroids. Therefore, SFCT is a biomarker that can be used to monitor the response to treatment. Previous studies have reported significant increases in the SFCT during exudative recurrences of DME [36]. In addition, a thickened choroid after anti-VEGF treatment may be interpreted as an indicator of a low response to anti-VEGF treatment and may signify the chronicity of macular edema or the predominance of the inflammatory component [40].

Our results indicate that it is not possible to use the baseline SFCT as a predictive parameter of the DME response to intravitreal treatment with anti-VEGF or corticosteroids. The decrease in CMT after treatment with aflibercept or dexamethasone did not correlate significantly with the SFCT values before intravitreal therapy. One possible explanation is that the state of the choroid in diabetic patients is highly variable, depending on the stage of DR, so its baseline central thickness before treatment could be found to be increased or decreased depending on whether an inflammatory or ischemic pathogenic mechanism prevails, respectively. Therefore, the value of SFCT as a biomarker acts more as a parameter in monitoring the response of DME to intravitreal treatment based on its variation than as a predictor of a possible good or bad response to treatment.

When comparing the variations in choroidal and macular thicknesses between the two treatment groups, the improvements were statistically superior in the dexamethasone-treated group. The mean reductions in CMT and SFCT after intravitreal treatment with aflibercept were 18.8 ± 14.7% and 13.8 ± 13.1%, while the mean reductions in CMT and SFCT after intravitreal treatment with dexamethasone were 29.7 ± 16.9% and 19.5 ± 9.6%. Intravitreal treatment with dexamethasone improved the macular thickness to levels comparable with eyes without DME, while aflibercept achieved macular thinning but without reaching the values of normal eyes without DME.

The secondary variables of age and gender were not shown to influence choroidal or macular thinning. Similarly, variations in both parameters after treatment with aflibercept or dexamethasone did not differ between eyes with and without previous intravitreal treatments. Patients who had received an intravitreal therapy other than the one studied in the last 6 months were excluded in order to exclusively assess the effect of the therapy under study [41]. The duration of diabetes was also not an important factor in choroidal or macular thinning after treatment, although there was a discrete trend towards greater reductions in eyes with shorter durations of diabetes. However, these data should be interpreted circumspectly, given the small number of eyes with diabetes durations of more than 30 years included in the sample.

In the control group of untreated eyes, there were no significant differences in the mean choroidal or macular values at baseline or at the end of the follow-up period.

The limitations of this work are those inherent to retrospective studies. Although the populations of the two study groups were homogeneous, the individual eyes presented different conditions. They differed, for example, in the number of previous injections and the history of eye surgery, although eyes operated on within the last two years were excluded. It has been shown that choroidal thickness may decrease after panretinal photocoagulation [42]. To minimize this possible interference, patients who had received laser treatment within the last two years were excluded. On the other hand, physiological diurnal variations of SFCT of up to 33.7 μm have been described [43,44]. Another limitation of the present study is that functional parameters such as visual acuity or contrast sensitivity, which are altered in patients with DME, were not assessed.

To conclude, an intravitreal treatment with anti-VEGF or corticosteroids in type 2 diabetic patients with macular edema decreases choroidal and macular thicknesses. According to our results, the SFCT is not a good biomarker of response (predictive biomarker), but it can be useful to assess the response to treatment, together with the CMT, and can be used in monitoring the response to DME treatment. 

## Figures and Tables

**Figure 1 jcm-12-00348-f001:**
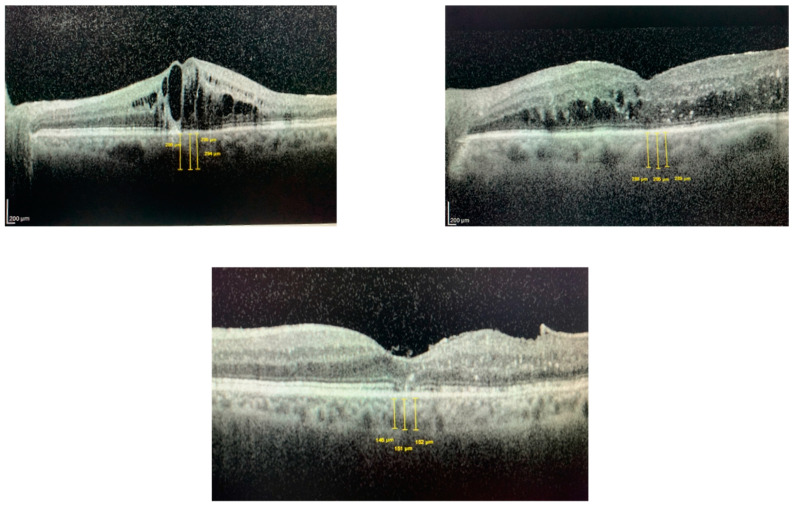
Three examples of manual measurements of the subfoveal choroidal thickness (SFCT) measurements performed with the OCT. Images from the study.

**Figure 2 jcm-12-00348-f002:**
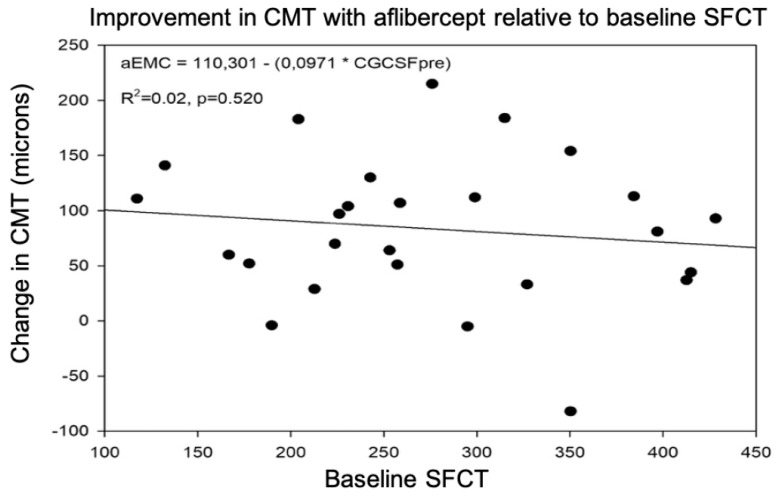
Changes in central macular thickness (CMT) related to baseline subfoveal choroidal thickness (SFCT) after intravitreal therapy with aflibercept.

**Figure 3 jcm-12-00348-f003:**
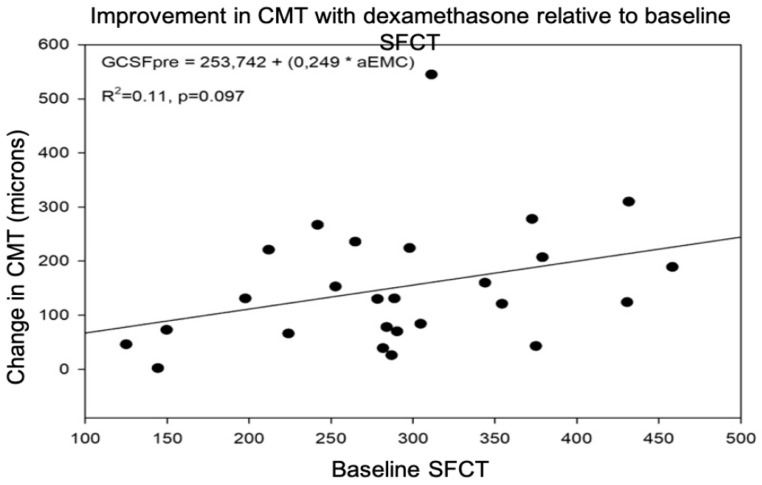
Changes in central macular thickness (CMT) related to baseline subfoveal choroidal thickness (SFCT) after intravitreal therapy with a dexamethasone implant.

**Figure 4 jcm-12-00348-f004:**
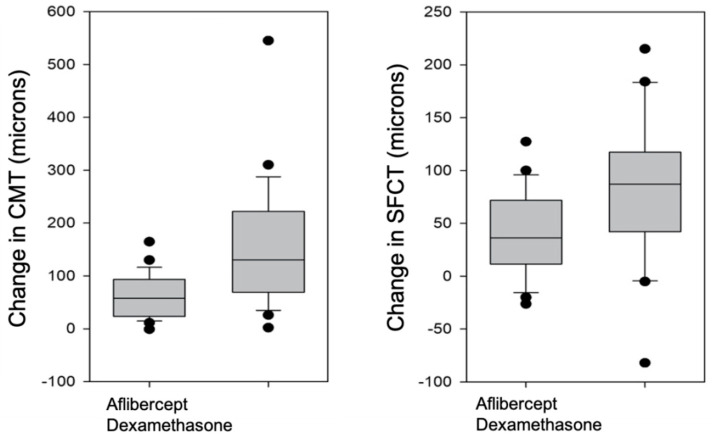
Change in central macular thickness (CMT) (left) and subfoveal choroidal thickness (SFCT) (right) after intravitreal therapy with aflibercept or dexamethasone.

**Table 1 jcm-12-00348-t001:** Results of treatment with aflibercept and dexamethasone implants.

	CMT Aflibercept	SFCT AFLIBERCEPT	CMT Dexamethasone Implant	SFCT Dexamethasone Implant	CMT Reduction Aflibercept versus Dexamethasone
Pre-treatment	441.4 ± 75.9 μm	274.7 ± 88.8 μm	480.1 ± 97.3 μm	291.6 ± 87.1 μm	
Post-treatment	357.8 ± 87.2 μm	235.6 ± 83.4 μm	328.0 ± 72.2 μm	229.8 ± 58.8 μm	
*p*-value	*p* < 0.001	*p* < 0.001	*p* < 0.001	*p* < 0.001	*p* = 0.022

CMT: Central macular Thickness; SFCT: Subfoveal choroidal Thickness.

## Data Availability

Data sharing not applicable.

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
