# Peer review of "Influence of Intravitreal Therapy on Choroidal Thickness in Patients with Diabetic Macular Edema"

_jcm, 2023, doi:10.3390/jcm12010348_

Round 1
Reviewer 1 Report
Dear Authors,
I wish to submit my review of the article titled: "Influence of Intravitreal Therapy on Choroidal Thickness in Patients with Diabetic Macular Edema."
The authors analyze an interesting topic that has already been discussed in other published papers. Nonetheless, the statistical analysis is well performed and profoundly analyzes the presented results. However, The article is often hard to follow and requires significant English language proofreading.
To be specific,
Abstract: Please report the p-value of the values reported in Methods (lines 24-26, page 1). Conclusions should be reported more clearly
Methods: Please define the study design clearly. Could you please define the OCT scan protocol used?
Results: Could you please summarize your results in tables?
Figure 1: Please define the Acronym.
Discussion: The discussion requires focusing more on the clinical relevance of the presented findings.
Text: Please proofread the English Language. Some acronyms are not specified, making the paper difficult to follow.
References:
Repeated References (18 and 32): Regatieri CV, Branchini L, Carmody J, Fujimoto JG, Duker JS. Choroidal thickness in patients with diabetic retinopathy analyzed by spectral-domain optical coherence tomography. Retina. 2012;32(3):563-568
Repeated Reference (19 and 33): Querques G, Lattanzio R, Querques L, del Turco C, Forte R, Pierro L, et al. Enhanced depth imaging optical coherence tomography in type 2 diabetes. Invest Ophthalmol Vis Sci. 2012;53(10):6017-6024.
Repeated Reference: (21-35): Wang W, Liu S, Qiu Z, He M, Wang L, Li Y, et al. Choroidal thickness in diabetes and diabetic retinopathy: A swept 483 source OCT study. Invest Ophthalmol Vis Sci. 2020;61(4):29.
Repeated Reference: (22-36): Hua R, Liu L, Wang X, Chen L. Imaging evidence of diabetic choroidopathy in vivo: Angiographic pathoanatomy and choroidal-enhanced depth imaging. PLoS One. 2013;8(12):1-5.
Repeated Reference: (23-37): Kim JT, Lee DH., Joe SG., Kim JG., Yoon YH. Changes in choroidal thickness in relation to the severity of retinopathy and macular edema in type 2 diabetic patients. Retina. 2013;54(3):3378-3384.
Repeated Reference: (38-48):Sen S, Ramasamy K, Sivaprasad S. Indicators of visual prognosis in diabetic macular oedema. J Pers Med. [Internet]. 515 2021;11(449):1-12. Disponible en: https://doi.org/10.3390/jpm11060449
Repeated Reference: (39-49)Campos A, Campos EJ, do Carmo A, Patrício M, Castro de Sousa JP, Ambrósio A, et al. Choroidal thickness changes 491 stratified by outcome in real-world treatment of diabetic macular edema. Graefe’s Arch Clin Exp Ophthalmol. 492 2018;256(10):1857-1865
Author Response
Thank you very much for reviewing the manuscript and for the comments. According to your recommendations:
1. The language has been revised by Jacqueline Van Velde.
2. References have been adjusted.
3. The p-values have been added to the abstract.
4. Study design has been redefined and the OCT scan used has been specified.
5. We have added some table for the results
6: The acronym in figure 1 has been defined.
Reviewer 2 Report
There are all along the manuscript some abbreviations must be defined namely: GCSF; MCE; SCMS and EMC.
Author Response
Thank you very much for proofreading the manuscript; there were indeed mistakes with some abbreviations which have been corrected.
Reviewer 3 Report
The use of biomarkers is becoming more and more clinically useful to early detect abnormalities or changes in anatomical retinal profile. In the course of diabetes, many structural changes occur, and any of them bring about harm on retinal tissues and drop in visual function.
This paper well analyzes changes in CMT and SFCT, after treatment with aflibercept and dexamethasone. The statistical analysis has been precisely carried out and graphs are clear and intelligible. Authors stressed the relative importance of SFCT and the fact that this biomarker is important but not essential to investigate the pathophysiological history of treated macula edema.
The reference list is updated. The authors could shorten discussion just a little bit in order to make the paper more fluent to readers. I appreciated the last paragraph where SFCT has been regarded not as a good predictor but as a good biomarker of response to intravitreal therapy.
Author Response
Thank you very much for reviewing the manuscript; the discussion has been revised and superfluous or non-essential information has been removed to make it shorter and easier to follow.
Round 2
Reviewer 1 Report
Dear Authors,
I wish to submit my review of the article you have revised.
The Authors improved the article and should be commended for their work. However, Some points still require proofreading.
Indeed, despite the great effort, English proofreading did not involve the whole paper (for example, the Introduction is as in the previous manuscript). Therefore, Please revise it to enhance language fluency.
"all eyes with availability of volume scan with follow up performed in the same OCT SPECTRALIS® (Heidelberg Engineering, Heidelberg, Germany) with sufficient quality to measure morphological changes in the retina and choroid at baseline and at the end of the study period". Please clarify this sentence. Could you please define the OCT Scan Protocol?
Discussion: The discussion was shortened. However, no statements on the clinical relevance of the presented findings were reported. Please add them.
Finally, The paper lack subheadings such as Author contributions or acknowledgments.
Author Response
Thank you again for reviewing the article. In response to comments and suggestions
- We have revised the introduction in both structure and language and believe it is better.
- Related to the OCT protocol and to clarify we wanted to say that only those patients who had OCT with follow-up were included in the study to make the results more objective and who had the same OCT protocol for the same reason; we have specifically defined the scan protocol (volume scan (high speed resolution and 20º x 20º scan pattern)).
- We have also added acknowledgements and contributions from the authors.
- Finally, we conclude that CSFT measurement can be useful to assess follow-up or response to treatment afterwards but is not useful as a predictive biomarker before starting intravitreal therapy.
